# Implementation of DDS Cloud Platform for Real-Time Data Acquisition of Sensors for a Legacy Machine

**Min-Huang Ho [1], Ming-Yi Lai [2] and Yung-Tien Liu [2,\*]**

[1]   Cross College Elite Tech Program, National Kaohsiung University of Science and Technology, Kaohsiung 82445, Taiwan; minhuang@nkust.edu.tw
[2]   Department of Mechatronics Engineering, National Kaohsiung University of Science and Technology, Kaohsiung 82445, Taiwan; f108103107@nkust.edu.tw
\*   Correspondence: ytliu@nkust.edu.tw

**Abstract:** Industry 4.0 (I4.0) is a multidisciplinary engineering principle combing the IoT (Internet of things), big data, and cloud computing to cope with the dynamic changing industry. In this paper, the DDS (data distribution service) communication protocol was employed to implement a cloud platform for data acquisition from various sensors on a precision legacy machine tool including an accelerometer and sound, temperature, brightness, and humidity sensors. The sensor signals were acquired using Raspberry Pi as the edge device, then published to the cloud using the DDS application, and stored in the MySQL database. Using the Django web server, the acquired sensor signals could be shown in real time on the webpage via a combination of MQTT and Node-RED. In addition, the motion displacement of the machine tool detected by the encoder could be recorded through the edge device for further performance examination. With the proposed DDS cloud platform, it is demonstrated that a legacy machine can enable sensing and communication abilities such that the development of a smart machine is achievable for future I4.0 application.

**Keywords:** DDS; MQTT; IoT; legacy machine; smart machine

## 1. Introduction

Current merchandized products such as computer, consumer, and communication (3C) products are increasingly being geared toward high performance, high precision, and multiple functions, but with small batch production and a short life cycle. Conventional mass production with a long leading time for developing a new product is no longer suitable for the dynamic changing industry. To cope well with the future trend in manufacturing, the Internet of things (IoT), driven in Germany by the so-called platform "Industry 4.0 (I4.0)", is believed to be a total solution for the dynamically changing industry. I4.0 focuses on product development and production scenarios. Consequently, during the production process, it is necessary to describe how manufacturing machinery and field devices are configured and how they function [1]. According to the reference architecture of a reported three-dimensional model of I4.0, the vertical axis "layers" represents the various perspectives, such as data maps, functional descriptions, communications behavior, hardware/assets, or business processes. The second axis represents the "life cycle and value stream" of products, machines, factories, etc. Lastly, the third axis of "hierarchy levels" describes the functional classification of various circumstances in which an intelligent field device, e.g., a smart sensor, is expectedly required. Therefore, to implement an I4.0 platform, one of the key components is to equip smart machines with sensing and communication abilities. In factories, however, most legacy machines were equipped before the launch of I4.0 production; thus, introducing additional sensing and communication abilities into these machines has raised great interest in practical applications.

To enable communication ability among various brands of legacy machines, a common communication protocol is required. A suitable protocol can improve the overall

flexibility and adaptability of the production line equipped with different smart machines. Regarding the communication protocols, a number of studies can be found in literature. Al-Fuqaha et al. presented an overview of the IoT with emphasis on enabling technologies, protocols, and application issues [2]. The global view of IoT applications was addressed, and the need for better horizontal integration among IoT services was presented. The OPC (Open Platform Communications) Foundation promoted that the OPC UA (Unified Architecture) could serve as the common data connectivity and collaboration standard for local and remote device access in IoT, M2M (machine-to-machine), and I4.0 settings [3,4]. On the other hand, Saxena et al. proposed a DDS-based communication framework for voltage regulation of smart distribution grids [5]. Endeley et al. presented a smart gateway that could enable interoperability between the OPC UA and DDS protocols. Profanter et al. compared the transmission performances among different protocols such as OPC UA, DDS, ROS (robot operating system), and MQTT (message queuing telemetry transport) [6]. In addition, Trunzer et al. proposed five characteristic indices of horizontal scalability (R1), adaptability or availability (R2), soft real-time capability (R3), quality of service (QoS) (R4), and configurability (R5) for evaluating the OPC UA and DDS protocols [7], in which the DDS featured comparable performance to the OPC UA but with higher configurability (R5). Ioana et al. proposed a solution to evaluate the real-time coexistence of OPC UA and DDS protocols, functioning in parallel and in a gateway context. Through the study, the compatibility and feasibility of the two protocols was confirmed [8]. Recently, Habib et al. proposed a middleware module incorporating the OPC UA and Representational State Transfer (REST) paradigms. RESTful web and cloud platforms were implemented to collect the middleware data, provide remote application support, and enable aggregated resource allocation in a database serve [9]. In addition, Vaclavova et al. proposed an IIoT (Industrial IoT) device solution according to the I4.0 concept, taking advantage of the benefits arising from OPC UA [10]. From the literature, it can be found that the OPC UA has high applicability in implementing the I4.0 platform, whereas many emerging real-time oriented protocols such as the DDS and MQTT are of interest.

In addition to communication protocols, sensors are the fundamental components to develop a smart machine in the factory. Javaid et al. discussed various sensors and their types, along with significant capabilities for manufacturing [11]. A total of 13 significant applications of sensors for I4.0 were summarized. Namjoshi and Rawat described the five major components of I4.0, in which smart monitoring based on sensors was considered essential [12]. The sensor types, the use of wireless sensor networks (WSNs) in preventative maintenance applications [13], and powering the sensor through the energy-harvesting technique for I4.0 are research topics that have drawn much interest recently [14]. In this study, environmental sensors and motion-detecting sensors were used to demonstrate the effectiveness of proposed DDS cloud platform.

In a previous study [15], the DDS communication protocol was employed to implement a cloud platform for data acquisition from various sensors. The implemented platform was briefly addressed, and the function of real-time data acquisition was examined. In this paper, the sensing information of various sensors mounted to a legacy machine, to determine its motion and environmental conditions, is reported via the application of Raspberry Pi. In addition, by employing the MQTT protocol together with the Node-RED programming tool, real-time monitoring of the sensed information is presented. Machine condition monitoring is essential in intelligent production for machine learning, assessment of machine condition, and fault diagnosis [16].

The remainder of this paper is organized as follows: Section 2 briefly introduces background information regarding the sensing process of the machine tool, protocols and programming tools, and web-based databases. This is followed by a description of the platform design in Section 3. In Section 4, details of system implementation and results are presented. Lastly, a conclusion is drawn in Section 5.

## 2. Sensing Information and Protocols

### 2.1. Acquisition of Motion Behavior of Machine Tool

Due to technical improvements, most current merchandized machine tools feature communication ability based on a popular protocol such as OPC UA. In factories, however, there are still many legacy machines equipped without such a function. To sense the motion behavior of a machine tool, an auxiliary position acquisition device (PAD) is usually required [17,18]. Figure 1 presents a sensing process of the motion behavior for a legacy machine tool. A PAD capable of capturing the position information of a machine tool is implemented via an additional shunt circuit connected to the encoder circuit. A FPGA (field-programmable gate array)-based RT controller is used to decode the encoder signals and provide the analog output (AO) voltage representing the motion, which is then recorded by a computer via an AD (analog-to-digital) converter. In this study, instead of the computer, the motion behavior was monitored in real time through a web server via the DDS protocol and stored on the server as historical data.

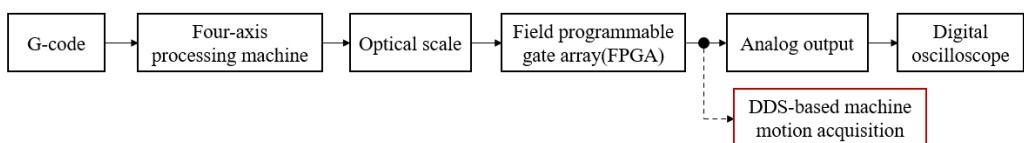

**Figure 1.** Sensing process of the motion displacement for a legacy machine tool.

### 2.2. DDS Protocol

DDS is a decentralized messaging protocol. It is an open international middleware standard which is maintained by the OMG (Object Management Group) directly addressing publish–subscribe communications for real-time and embedded systems. It uses the concept of a virtual global data space or domain for data sharing. The DDS domain is implemented in an XML file and shared with various DDS applications, which can communicate with each other if they share the same domain. Every node in the domain can publish and subscribe a specific topic for message exchange. During the exchanging process, the node and its topic can be specified by their own QoS (quality of service) according to the entities and characteristics of the message or data. Through the use of the QoS with respect to bandwidth, priority, delivery deadlines, reliability, and data volatility, we can alter the attribute for message transmission. Figure 2 depicts a typical diagram for the concept of message exchanges via the DDS protocol [19].

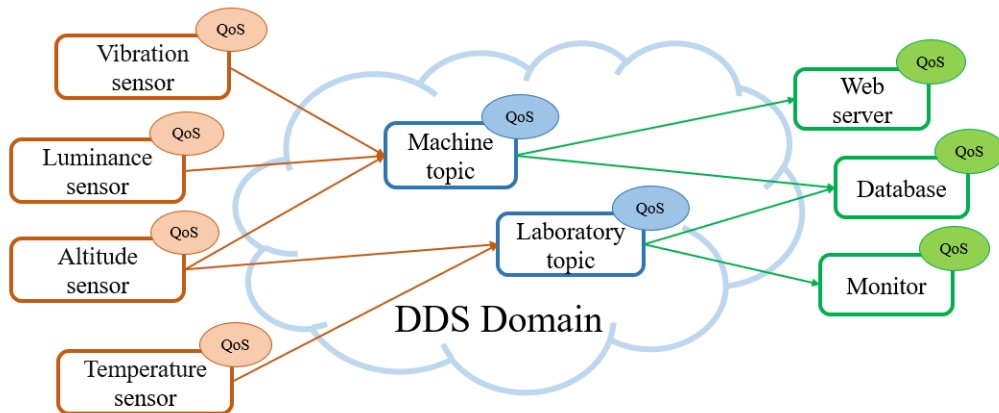

**Figure 2.** The logic of DDS protocol for messaging [19].

### 2.3. MQTT and Node-RED

MQTT is a lightweight protocol based on the publish and subscribe model, and it allows one-to-one, one-to-many, and many-to-many connections. The MQTT protocol

requires a server acting as the broker to transmit the information with specified topics from the publishers to subscribers.

Node-RED is an IBM Inc. contributed open-source JS Foundation project. It is a graphical programming tool for developing an application service visually expressing the information of hardware on the web. Using the provided user interface (UI) of Node-RED, the developer can design a suitable dashboard for a sensor via MQTT such that the information of the sensor can be shown in real time on the web.

In this study, the tool of "MQTT in" provided by Node-RED was employed to establish an input data stream with a specified topic from the local broker of MQTT on the webserver. Once the broker receives the sensed information from DDS application via MQTT, the information can be redirected to the Node-RED dashboard UI component and be shown in real time on the web.

### 2.4. Characteristics of the Proposed DDS Platform

In the proposed DDS-based platform, the DDS applications are executed on both the edge device and the web server in the same DDS domain. To show the acquired information in real time, the MQTT broker is implemented in the web server such that the DDS can receive the data and transmit it to the Node-RED UI via MQTT. Benefiting from the advantages of DDS in terms of decentralized and real-time characteristics, the proposed system is flexible, distributable, and reconfigurable. That is, a sensor module can be easily joined with the platform or removed from the platform. With high robustness of information sharing in the DDS domain, the proposed DDS platform is suitable for application to production lines and factories for monitoring the sensors in real time.

## 3. Platform Design

### 3.1. Problem Definition

The scenario of data acquisition for various sensors is shown in Figure 3, where three different locations of "machine", "laboratory #1", and "laboratory #2" are presented. Each location might have specific sensors for monitoring environmental or operational conditions. In many cases, we need to dynamically add or remove the sensing locations. How to collect the sensed information obtained from different locations is a fundamental task to meet the requirement of I4.0 production. The target of this study was, therefore, to implement a DDS-based platform to collect the sensed information.

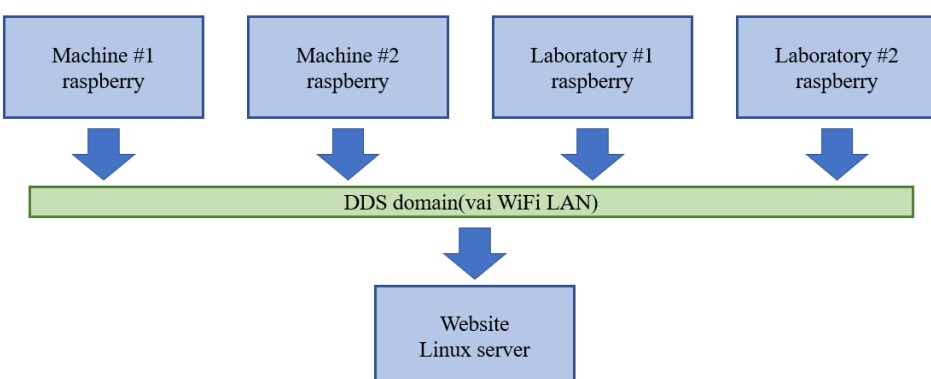

**Figure 3.** Data acquisition for different locations.

Figure 4 shows the subject flow diagram of the platform having four topics of "machine sensor", "laboratory #1 sensor", "laboratory #2 sensor", and "motion", which are acquired from three locations. Each location is equipped with an edge device of Raspberry Pi for transmitting the sensed information with a specific topic and sensor type to the webserver. Then, the webserver shows the sensed information on the web and stores the information to the database as historical data for further application.

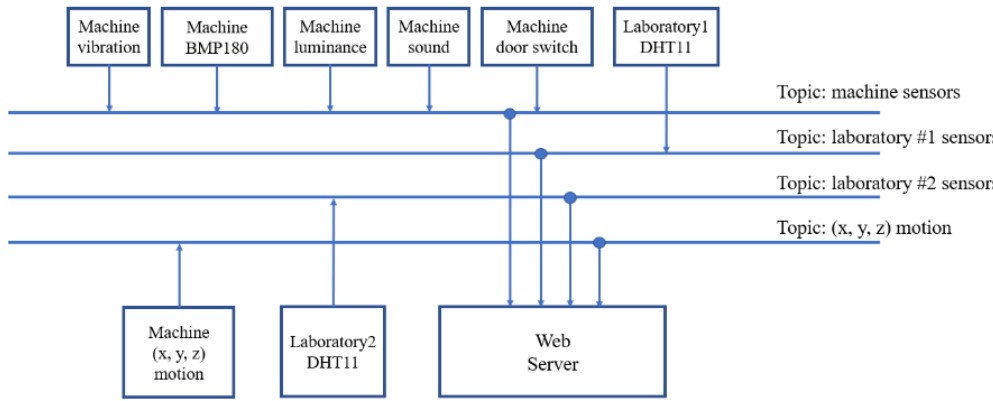

**Figure 4.** Subject flow diagram.

### 3.2. Proposed Platform Architecture

Figure 5 shows the proposed cloud platform mainly consisting of the edge device, DDS domain, database, and monitoring web server with a website. The dashboard in the web server is implemented using the Django web framework and Node-RED. The website provides a link to the Node-RED dashboard UI for real-time monitoring of the data from the edge devices. It also provides an interface for querying the historical data stored in the database.

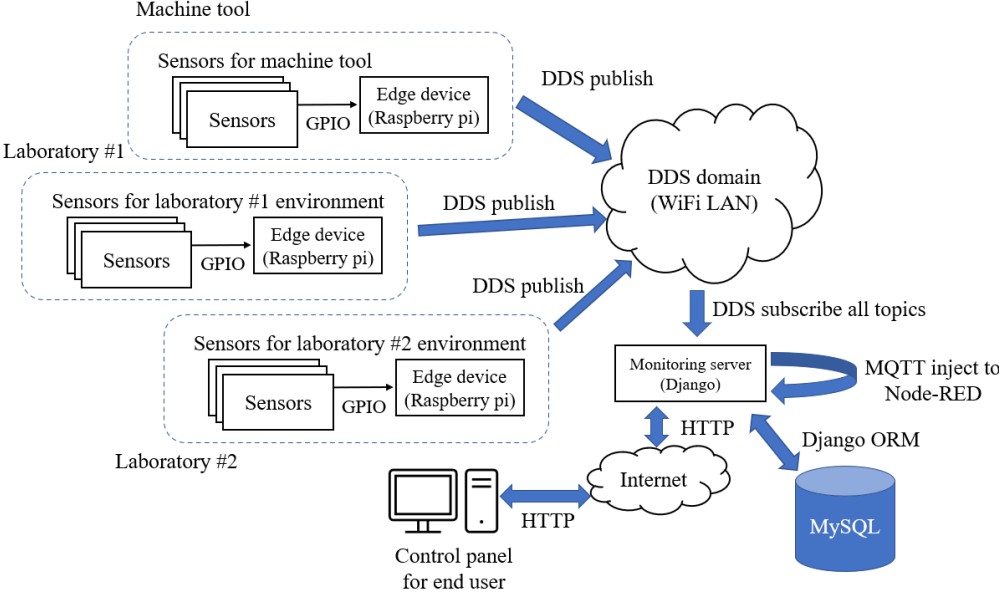

**Figure 5.** Architecture of the proposed cloud platform.

As shown in Figure 5, the sensor signals are connected to the GPIO (general purpose input/output) ports of edge device of Raspberry Pi for data collection. The program of the DDS application implemented by the OS of Raspberry Pi can read a written XML (extensible markup language) file to transmit the topic format with bundled QoS configuration, and then publish the information with the RTI Connext DDS. The published information contains the sensed data together with a location, i.e., "machine", "laboratory #1", or "laboratory #2".

In the web server, the program of the DDS application is implemented to subscribe all possible topics from edge devices. Moreover, the program can filter the interested data, store the data in a database via Django ORM (object relational mapper), and publish the data to specified topics via the MQTT protocol.

To show the received data on the web page, the web server has to set up a MQTT broker and activate the Node-RED framework service. Several sensor UI components are laid out to subscribe specified MQTT topics. Once the data are published with a specified topic, the value is reflected in real time on the appropriate sensor UI component of the web page.

### 3.3. Raspberry Pi-Based Sensor Publisher

Since the DDS is a distributed publish/subscribe model, every edge device has its own DDS application to publish the sensed data using the same XML file for topic and QoS configuration. The edge devices can be freely joined or removed from the DDS domain. Once joined, the DDS application executed on the edge device can acquire the data from the sensors, filter the data, and then publish the filtered data to the DDS domain with specific topics. In the proposed DDS platform, all the edge devices connected to the wireless LAN are not accessible from the Internet.

### 3.4. Messaging between Raspberry Pi and Server via DDS

As presented in this study, the platform has more than one edge device using Raspberry Pi 4. Each Raspberry Pi runs the DDS application serving as the sensor publish agent for publishing the acquired data, and the web server executes another DDS application serving as a subscribe agent for subscription.

In the proposed system, several programs of subscribe agents are run in the background of the web server. The subscribe agent of the web server is another DDS application for receiving the data with the specified DDS topics. The received information is then stored into MySQL databases via Django ORM, and the web charts in Node-RED UI page are updated accordingly.

### 3.5. Display of Real-Time Chart with MQTT and Node-RED

Node-RED has an "MQTT in" component to subscribe to a specified MQTT topic. The component can be connected to a Node-RED dashboard sensor component to show the sensed value with a chart or numerical number. Utilizing this function, the sensed value from a DDS topic can be displayed in real time. Therefore, all we have to do is publish the value to a specified MQTT topic. Figure 6 shows part of the Node-RED flow of the implemented system.

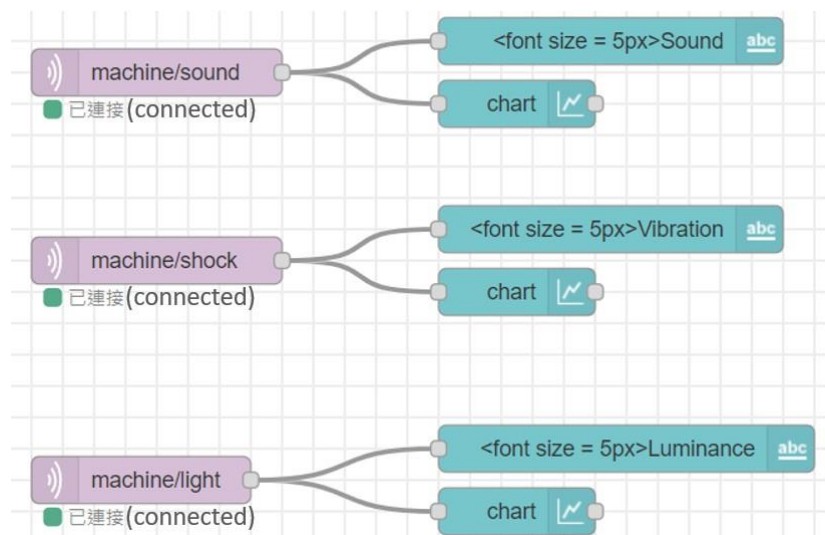

**Figure 6.** Part of the Node-RED flow of the implemented system.

## 4. System Implementation and Results

### 4.1. Implemented Platform

Figure 7 shows the data flow of the implemented cloud platform including the sensors' input signals, the transmission protocols of DDS, MQTT, MySQL database, and browser. The DDS framework used in the proposed system is the RTI Connext. We utilize Raspberry Pi 4's I/O abilities to collect the data from the sensors, and we publish the data with specified topics via the DDS.

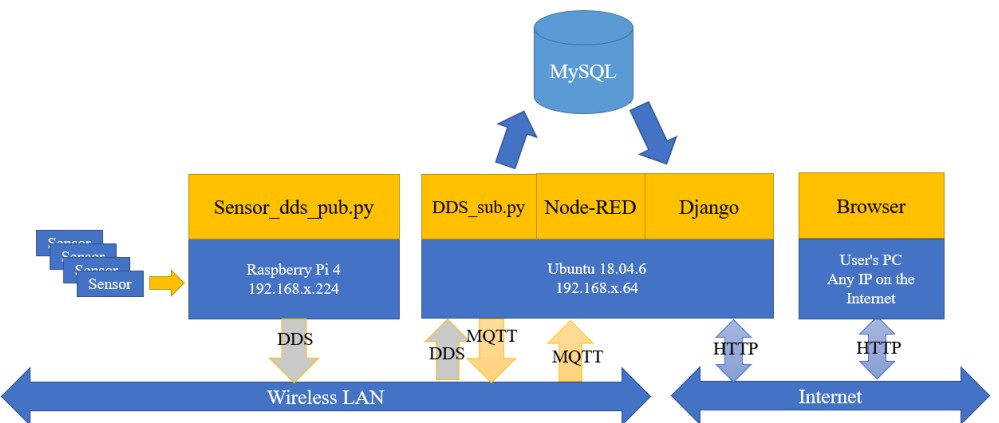

**Figure 7.** Data flow and main components of the implemented system.

### 4.2. Sensing Module

As shown in Figure 8, there are four edge devices to collect data from various sensors at three different locations. Every edge device can be composed of several sensors as an individual sensing module. The topic or sensing module of the "machine sensor" consisted of five types of sensors, i.e., "sound", "vibration", "luminance", "optical switch", and "barometric pressure", mounted on the machine tool. Since Raspberry Pi can receive digital signals only, an analog-to-digital (A/D) converter such as MCP 3008 is required for analog devices. In addition, since Raspberry Pi 4 has only one connector for the A/D converter, the motion of the machine tool is acquired by another Raspberry Pi 4. The other two sensors of "humility" and "barometric pressure" are separately located at laboratory #1 and #2.

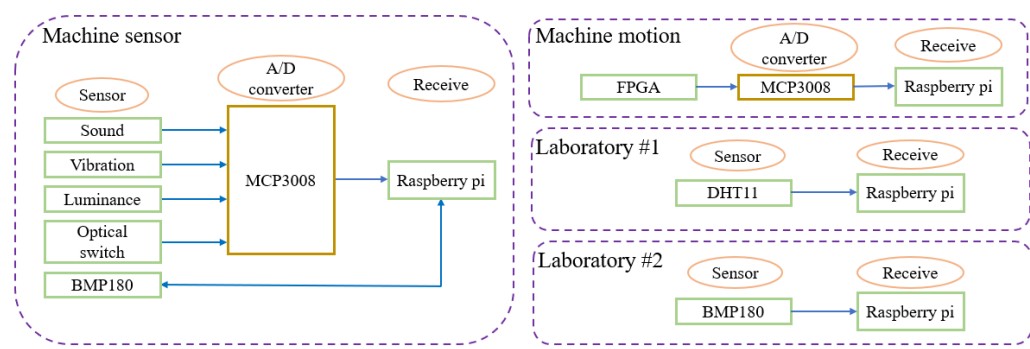

**Figure 8.** Physical connection of sensor and Raspberry Pi.

### 4.3. DDS Topic and Its Implementation

To implement the DDS applications, it is required to assign the type of topic and specify the related domain configurations, which are XML-based documents. Some of the settings in the sensor.xml are shown in Figure 9.

```
<types>
  <struct name="SensorType" extensibility="extensible">
    <member name="station" stringMaxLength="20" type="string" key="true"/>
    <member name="temperature" type="float64"/>
    <member name="humidity" type="float64"/>
    <member name="sound" type="long"/>
    <member name="light" type="long"/>
    <member name='acx' type="long"/>
     <member name='acy' type="long"/>
    <member name='acz' type="long"/>
  </struct>
</types>
```

**Figure 9.** The topic data type expressed in the sensor.xml.

As shown in the sensor.xml, the "station" is used to represent name of the edge device, which is expected to be integrated into a production line or a smart machine. The other expressions indicate the names of sensors and types of data. The XML file is used in Raspberry Pi to publish the sensor values and is used in the server's application for subscription. The QoS configuration is also specified in the same XML. In this study, we used the default QoS setting, as shown in Figure 10.

```
<qos_library name="QosLibrary">
        <qos_profile name="DefaultProfile"
                    base_name="BuiltinQosLib::Generic.StrictReliable"
                    is_default_qos="true">
            <domain_participant_qos>
                <participant_name>
                    <name>Sensor Type</name>
                </participant_name>
            </domain_participant_qos>
        </qos_profile>
    </qos_library>
```

**Figure 10.** Sample QoS configuration of the proposed platform.

### 4.4. Motion Displacement Expressed by AO Signal and Sensed via Raspberry Pi

As described in Section 2.1, an FPGA-based RT controller was used to decode the encoder signals and provide the AO voltage representing the motion displacement. Figure 11 shows the measured AO signal of the analog output via Raspberry Pi using the G-code command with a displacement of 1 mm and feeding speed of 90 mm/min.

### 4.5. Motion Displacement Directly Sensed via Raspberry Pi

In fact, the motion of the machine tool along every axis is sensed via an encoder with a resolution of 0.1 μm. The encoder can generate two pulse waveforms of PA and PB with a phase difference of 90° via a quadruple differential line receiver. When the encoder moves forward, PA indicates the phase advance; otherwise, PB indicates the phase advance, as shown in Figure 12. According to the waveforms, an algorithm was implemented in Raspberry Pi to count the number of pulse waveforms, as shown in Figure 13. As indicated in the algorithm, when either PA or PB is high, the number of counts is increased by 1 if PA = high or decreased by 1 if PA = low. When both signals are low, the number of counts remains at a fixed value. The RT controller can detect the encoder signals in real time.

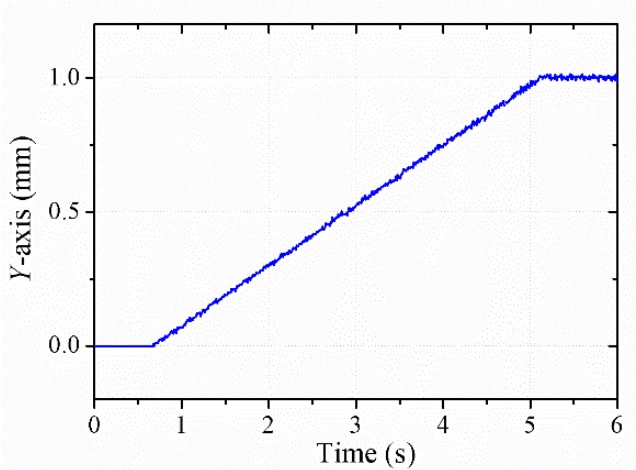

**Figure 11.** Motion behavior of machine tool acquired via Raspberry Pi.

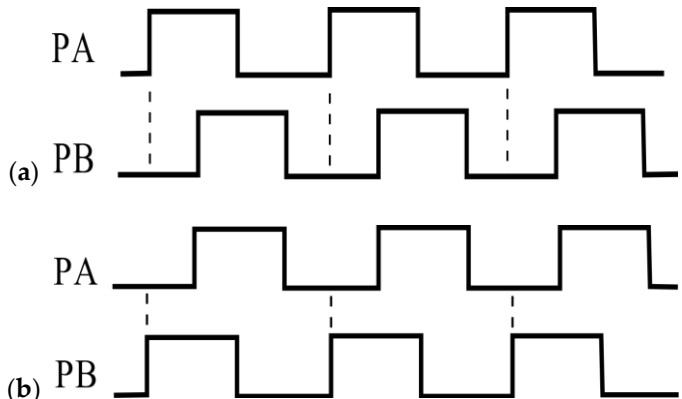

**Figure 12.** Pulse waveforms generated by the encoder: (**a**) forward motion; (**b**) backward motion.

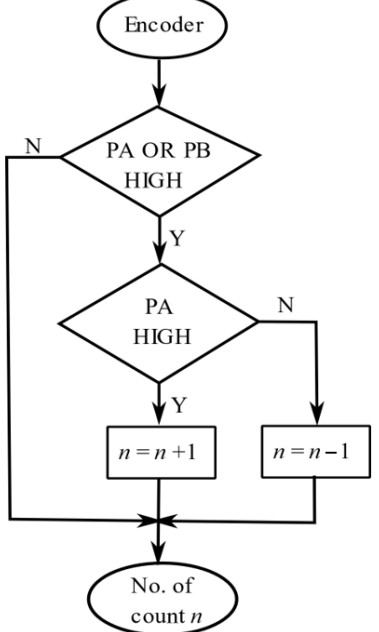

**Figure 13.** Algorithm for counting the number of pulse waveforms.

Figure 14 shows the measured results of motion displacement via Raspberry Pi according to the pulse counts. Three experiments of step motions with displacements of 1 mm, 5 mm, and 10 mm were performed on the basis of G-code commands. As shown in the results, the three measured results were well aligned. This indicates that a good measure repeatability was obtained.

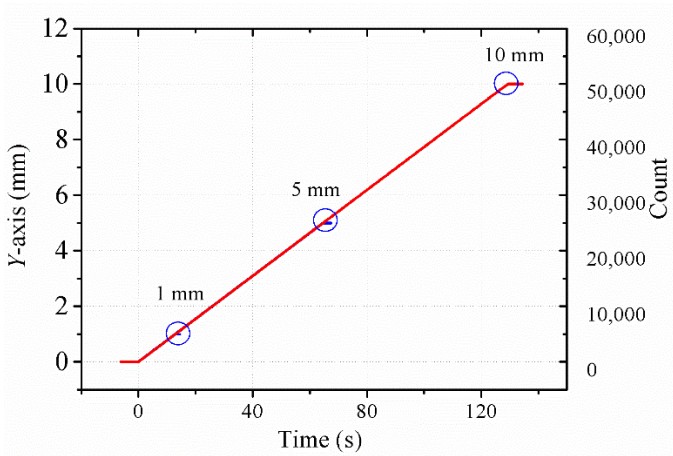

**Figure 14.** Motion behavior of machine tool acquired via Raspberry Pi.

### 4.6. Dashboard Implementation of Sensors

Figure 15 shows the real-time data displays on the web page. All the sensor signals acquired from different locations can be simultaneously shown with the link to the application of Node-RED. This allows the user to observe the operational condition of the legacy machine and the on-site environment in real time. In addition, the acquired data stored in the database can be retrieved and displayed on the web page as shown in Figure 16, where historical data of vibration and sound are presented. On the web page, a filter is used to select a specific acquired information for observation. Moreover, a Web API interface is provided to retrieve the stored data such that the machining performance of the legacy machine relating to the operational and environmental conditions can be precisely examined in a future study.

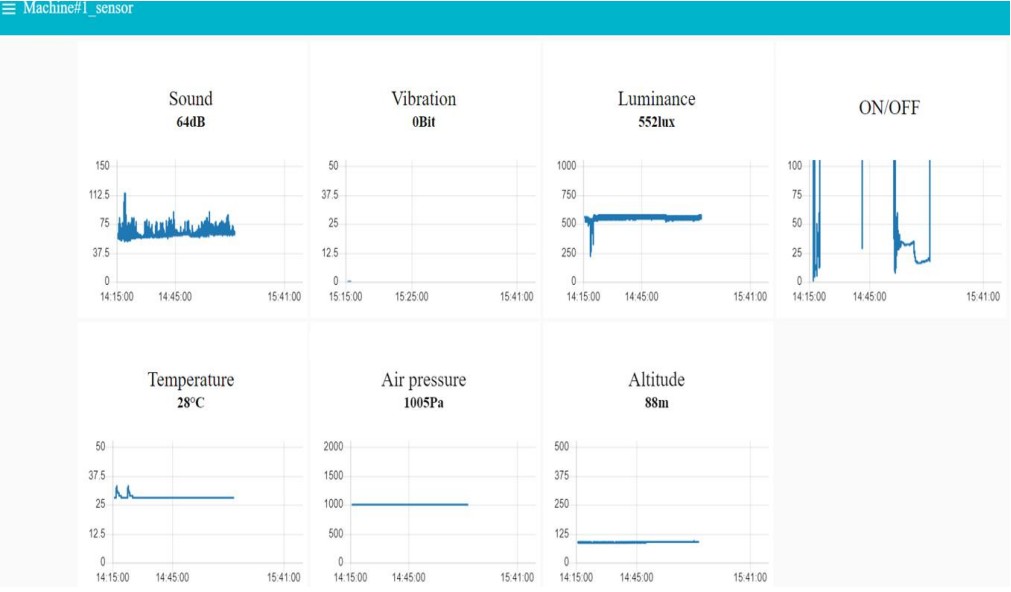

**Figure 15.** Real-time displays of sensor dashboards on the website.

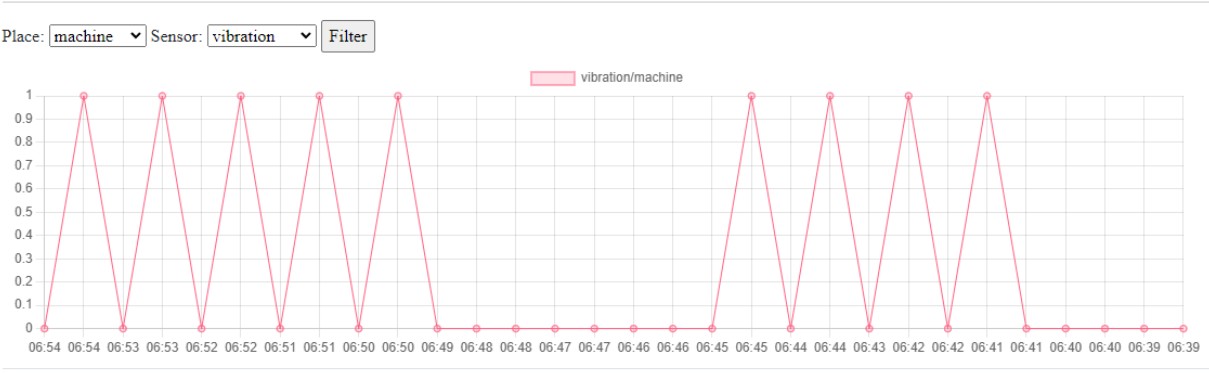

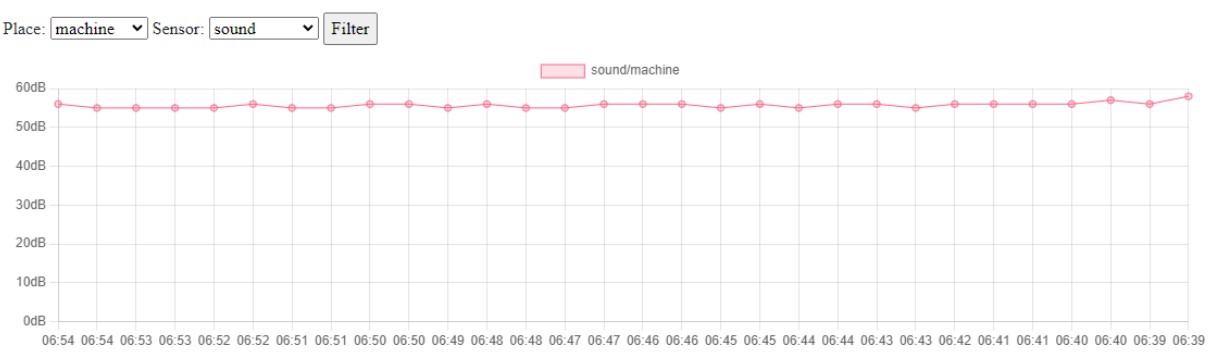

**Figure 16.** Historical data of measured signal: (**a**) vibration; (**b**) sound.

## 5. Conclusions

In this study, a cloud platform based on the DDS protocol was implemented for a legacy machine. Various sensors were used to demonstrate the function of real-time data acquisition. The main results are summarized as follows:

(1) The DDS-based platform can receive the sensor information, publish the data to a private topic, and store the information in a database.

(2) The sensed information of sensors including measured values and time history charts can be shown by Node-RED via the MQTT protocol.

(3) The motion displacement of the machine tool can be well acquired by the edge device of Raspberry PI.

(4) A database based on MySQL is implemented for the data retrieved.

According to the results obtained, the proposed DDS cloud platform can enable a legacy machine with sensing and communication abilities. Future studies can examine the

transmission performance of the platform, develop a cloud computing system to establish a smart machine based on the acquired data, and extend the application to machining sensors such as for cutting force, acoustic emission waves, and energy consumption.

**Author Contributions:** Conceptualization, M.-H.H. and Y.-T.L.; methodology, M.-H.H., M.-Y.L. and Y.-T.L.; software, M.-H.H., M.-Y.L. and Y.-T.L.; validation, M.-H.H., M.-Y.L. and Y.-T.L.; formal analysis, Y.-T.L.; investigation, M.-H.H. and M.-Y.L.; resources, M.-H.H. and Y.-T.L.; data curation, M.-H.H., M.-Y.L. and Y.-T.L.; writing—original draft preparation, M.-H.H., M.-Y.L. and Y.-T.L.; writing—review and editing, M.-H.H. and Y.-T.L.; visualization, M.-H.H. and Y.-T.L.; supervision, M.-H.H. and Y.-T.L.; project administration, Y.-T.L.; funding acquisition, M.-H.H. and Y.-T.L. All authors have read and agreed to the published version of the manuscript.

**Funding:** This research was funded by the Ministry of Science and Technology of the Republic of China (Taiwan) with grant number MOST 110-2221-E-992-067.

**Acknowledgments:** Financial support from the Ministry of Science and Technology of the Republic of China (Taiwan) with grant No. MOST 110-2221-E-992-067 is gratefully acknowledged.

**Conflicts of Interest:** The authors declare no conflict of interest.

## Abbreviations

| | |
|---|---|
| DDS | Data distribution service |
| Django ORM | Django object relational mapper |
| FPGA | Field-programmable gate array |
| GPIO | General purpose input/output |
| HTTP | Hypertext transfer protocol |
| I4.0 | Industry 4.0 |
| IIoT | Industrial IoT |
| IoT | Internet of things |
| LAN | Local area network |
| M2M | Machine-to-machine |
| MQTT | Message queuing telemetry transport |
| OPC | Open platform communications |
| OPC UA | OPC Unified Architecture |
| PAD | Position acquisition device |
| QoS | Quality of service |
| REST | Representational state transfer |
| RT controller | Real-time controller |
| UI | User interface |
| WAN | Wide area network |
| Web API | Web application programming interface |
| XML file | Extensible markup language file |

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
