# Peer review of "Implementation of DDS Cloud Platform for Real-Time Data Acquisition of Sensors for a Legacy Machine"

_electronics, doi:10.3390/electronics11132096_

Round 1

Reviewer 1 Report

Please, refer to the attached document! Thanks

Author Response

Dear Reviewer:

The authors wish to thank the Reviewers for the valuable evaluations and suggestions. The manuscript has been greatly improved due to reviewers’ suggestions. The revisions are summarized as follows:

Comments from Reviewer:

The manuscript entitled “Implementation of DDS Cloud Platform for Real-time Data Acquisition of Sensors for a Legacy Machine” has been reviewed. The authors proposed to implement a cloud platform based on the DDS protocol for data acquisition of various sensors on a precision legacy machine. The proposed paper is very well structured, the ideas are clear and the writing is concise and argumentative. The literature review is comprehensive and managed to successfully discuss the importance of your research, from both a theoretical and an applied perspective. The authors refer a good state of the art (references) to develop the theoretical part of their paper, although some further discussion about the energy needed to power supply the sensors should be evaluated in the specific section. Both research questions and hypotheses are derived and sustained by the literature review, and are pertinent to proposed study. The method choose is adequate and the authors providing arguments from previous studies. However, the paper presents some questions/problems/imperfections noticed and listed below for significant improvements. The article is attached and the imperfections highlighted in yellow.

(-) the text is reasonably clear so this reviewer recommends a minimum control of English language.

ANS:

      Thanks for the reviewer’s comment. The authors have made the best efforts on improving the readability of English expression.

(-) The abstract is very dispersive. This type of journals allows up to 200 words, which should be best used to make the article focused on the point. Although the aim of the paper is clear, in general, the abstract should be elegant and well written highlighting points as the novelty respect the state of art, and showing the results obtained. It is opinion of this reviewer that the abstract requires a restructuring to have a better fluency. It should be better organized.

ANS:

Thank you for the reviewer’s comment. The authors have made some modification relating to stress the originality of the paper as follows,

The Industry 4.0 (I4.0) is a multi-disciplinary engineering by combing the IoT (Internet-of-Things), big data, and cloud computing to cope well with the dynamic changing industry. In this paper, the DDS (Data distributed service) communication protocol was employed to implement a cloud platform for data acquisition of various sensors on a precision legacy machine tool including accelerometer, sound, temperature, brightness, and humidity, etc. In addition, the motion behavior of the machine tool could be recorded through the edge device. The sensor signals were acquired by the Raspberry Pi serving as the edge device and then published with the DDS application to the cloud and stored to the MySQL database. With the Django web server, the sensor signals acquired could be real-time shown on the webpage by the combination of MQTT and Node-RED. In addition, the motion displacement of the machine tool detected by the encoder could be recorded through the edge device for further performance examination. It is expected that With the proposed DDS cloud platform, it has been demonstrated that can be applied to a legacy machine could enable sensing and communication abilities such that the development of a smart machine is achievable for future I4.0 application.

(-) The keywords are appropriate. However, this is only a reviewer's opinion, it is better to use keywords that better highlight the focus of the paper. Authors have the possibility to insert a maximum of five keywords.

ANS:

The Keywords are modified as follows, à

Keywords: DDS; MQTT; IoT; Legacy Machine; Smart Machine Cloud Computing; Monitoring

(-) It is the opinion of this reviewer that the benefits of the cloud platform based on the DDS protocol compared to the current state of the art should be highlighted in the text

ANS:

  Some other benefits of the cloud platform are additionally in Section 2.4 as follows, à

  2.4 Characteristics of the proposed DDS platform

In the proposed DDS framework, the DDS applications are executed on both the edge device and web server in the same DDS domain. To real-time show the acquired information, the MQTT broker is implemented in the server such that the DDS can receive the data and transmit it to the Node-RED user-interface via the topic of MQTT. Benefiting from the advantages of DDS on decentralized and real-time characteristics, the proposed system features flexible, distributable, and reconfigurable. That is, a sensor module can be easily joined with the platform or removed from the platform. Having high robustness of information sharing in the DDS domain, the proposed DDS platform It is suitable for the applications to the production lines and factories for real-time monitoring the sensors.

(-) Figure 15 is unclear. I have to zoom up to 100% to see something. Please check it.

ANS:

The real-time data displays on the web page is modified as follows, à

Figure 15. Real-time displays of dashboard of sensors on the web site.

(-) In this paper, many abbreviations have been used. I suggest to the authors to make a section at the end of the paper with the title “Abbreviations”. All words and their abbreviations will be entered here. This will make it easy for the reader.

ANS.

Thanks for the comment. A section is additionally provided as follows, à

Abbreviations

DDS: Data distribution service

Django ORM: Django object relational mapper

FPGA: Field programmable gate array

GPIO: General purpose input/output

HTTP: Hyper text transfer protocol

I4.0: Industry 4.0

IIoT: Industrial IoT

IoT: Internet-of-things

LAN: Local area network

M2M: Machine-to-machine

MQTT: Message queuing telemetry transport

OPC: Open platform communications

OPC UA: OPC Unified architecture

PAD: Position acquisition device

QoS: Quality-of-service

REST: Representational state transfer

RT controller: Real-time controller

RTI Connext:

UI: User interface

WAN: Wide area network

Web API: Web application programming interface

XML file: Extensible markup language file

(-) Several imperfections are highlighted in yellow in the text. For example, replace the semicolon with a period. Never start the sentence with "End" (row 36).

ANS:

Thanks for the comment. All unsuitable semicolons have been replaced by periods. The sentence starting with “And” is revised as follows, à

…represents the “Life cycle and value stream” of products, machines, factories, etc. ; And Finally, the third axis is the “Hierarchy levels” describing functional classification of various circumstances…

(-) The Section 4.5 for this reviewer represents an important part of this article. In fact, it is a representation of the data obtained from the sensors. The observation is, a data analysis section would make clearer how you plan to use the data you are collecting to answer your research question and test your hypotheses. Authors should provide more details on this part.

ANS:

Thanks for valuable suggestion relating to the acquired sensor signals. The description is revised as follows, à

Figure 15 shows the real-time data displays on the web page. All the sensors signal acquired from different locations can be simultaneously shown with the link to the application of Node-RED. This allows the user to real-time observe the operational condition of the legacy machine and on-site environment. In addition, the acquired data stored in the database can be retrieved and displayed on the web page as shown in Figure 16, where two history data of vibration and temperature are presented. On the web page, a filter is used to select a specific acquired information for observation. Moreover, a Web API interface is provided to retrieve the stored data such that the machining performance of a legacy machine relating to the operational and environmental conditions can be precisely examined in future study.  

(-) Internet of things (IoT) is a revolutionizing technology based on an ecosystem of connected objects and embedded devices. However, batteries cannot power supply a network of IoT devices in particular if they are installed in hard to reach areas. A viable solution is to harvest energy from environment and then provide enough energy to the devices to perform their operations.

This reviewer suggests one possible publication only for the concept of how to power supply the sensors through the energy harvesting technique

- Citroni, R.; Di Paolo, F.; Livreri, P. Evaluation of an optical energy harvester for SHM application, AEU - International Journal of Electronics and Communications Volume 111, November 2019, 152918

ANS:

Thanks for the comment. A literature review relating to Sensor is additionally provided in Introduction as follows, à

In addition to communication protocols, sensors are the fundamental components to develop a smart machine in factory. Javaid et al. discuss various sensors and their types, along with significant capabilities for manufacturing [11]. A total of thirteen significant applications of sensors for I4.0 are summarized. Namjoshi and Rawat described the five major components of I4.0 in which the smart monitoring based on sensors is essential [12]. Not only the sensors types, how to use wireless sensor networks (WSNs) in preventative maintenance applications[13], and how to power the sensor through the energy harvesting technique for I4.0 drawn much interesting recently [14]. In this study, two types of environmental sensors and motion detecting sensor are used to demonstrate the effectiveness of proposed DDS cloud platform.  

(-) Finally, I think, that the authors in “conclusion section “should give their opinion on the future prospective about this new study and what could be its main advantage not only a list of final considerations.

ANS:

To stress the research results and future applications, modification is made in Conclusion as     follows, à

…According to the results obtained, it has been demonstrated that the proposed DDS cloud platform could enable a legacy machine with sensing and communication abilities. Future works are considered to examine the transmission performances of the platform, develop a cloud computing system to establish a smart machine based on the acquired data, and extend the actual applications to machining sensors such as for cutting force, acoustic emission wave, and energy consumption, etc.  legacy machines to enable communication abilities are expected.

I am sure that the paper has solid basis and the vision of the paper is very good but it needs to be properly described and detailed, showing how it will contribute to the advancement in IoT field. In light of these results, this reviewer suggests reviewing some points in this paper. For this purpose, a major review by taking into consideration also these comments before that this article can be published is needed.

ANS:

Thanks for the review’s positive evaluations. The authors believe that the manuscript has been greatly improved in accordance with the review comments.

Reviewer 2 Report

The paper is quite well prepared, however, it needs to be corrected.

Please consider:

1) Providing sources under the (figures).

2) Extending the bibliography

3) Elaborating Conclusions (future works are too shortly expressed)

4) Proofreading the paper and eliminating editorial errors, e.g.:

Line 10: Correct: Internet of Things
Line 11: Correct: Data Distributed Service
Line 35: Correct: processes.
Line 36: Correct: etc.
Line 46: Correct: literature.

etc.

Author Response

Dear Reviewer:

The authors wish to thank the Reviewers for the valuable evaluations and suggestions. The manuscript has been greatly improved due to reviewers’ suggestions. The revisions are summarized as follows:

Reviewer 2’s comments:

The paper is quite well prepared, however, it needs to be corrected.

ANS :

Thanks for the review’s positive evaluations. The authors believe that the manuscript has been greatly improved in accordance with the review comments.

Please consider:

  • Providing sources under the (figures).

ANS :

Thanks for the reminder of cited reference. Except Figure 2, all the figures are drawn based on authors’ original concepts. A cited reference is provided under the caption of Figure 2.

  • Extending the bibliography

ANS:

Thanks for the comment. A literature review relating to Sensor is additionally provided in Introduction as follows, à

In addition to communication protocols, sensors are the fundamental components to develop a smart machine in factory. Javaid et al. discuss various sensors and their types, along with significant capabilities for manufacturing [11]. A total of thirteen significant applications of sensors for I4.0 are summarized. Namjoshi and Rawat described the five major components of I4.0 in which the smart monitoring based on sensors is essential [12]. Not only the sensors types, how to use wireless sensor networks (WSNs) in preventative maintenance applications [13], and how to power the sensor through the energy harvesting technique for I4.0 drawn much interesting recently [14]. In this study, two types of environmental sensors and motion detecting sensor are used to demonstrate the effectiveness of proposed DDS cloud platform.    

  • Elaborating Conclusions (future works are too shortly expressed)

ANS:

To stress the research results and future applications, modification is made in Conclusion as     follows, à

…According to the results obtained, it has been demonstrated that the proposed DDS cloud platform could enable a legacy machine with sensing and communication abilities. Future works are considered to examine the transmission performances of the platform, develop a cloud computing system to establish a smart machine based on the acquired data, and extend the actual applications to machining sensors such as for cutting force, acoustic emission wave, and energy consumption, etc.  legacy machines to enable communication abilities are expected.

4) Proofreading the paper and eliminating editorial errors, e.g.:

Line 10: Correct: Internet of Things
Line 11: Correct: Data Distributed Service
Line 35: Correct: processes.
Line 36: Correct: etc.
Line 46: Correct: literature.

etc.

  ANS:

Thanks for the review’s comments. The authors have made the best efforts to proofread the   manuscript with error free.

Round 2

Reviewer 1 Report

 Please, refer to the attached document. Thanks!
